# Kidney Transplant Recipients with Acute Antibody-Mediated Rejection Show Altered Levels of Matrix Metalloproteinases and Their Inhibitors: Evaluation of Circulating MMP and TIMP Profiles

**DOI:** 10.3390/ijms26136011

**Published:** 2025-06-23

**Authors:** Miguel A. Vázquez-Toledo, Fausto Sánchez-Muñoz, Iván Zepeda-Quiroz, Carlos A. Guzmán-Martín, Horacio Osorio-Alonso, Juárez-Villa Daniel, Ma. Virgilia Soto-Abraham, Bernardo Moguel-González, Rommel Chacón-Salinas, César Flores-Gama, Rashidi Springall

**Affiliations:** 1Posgrado en Ciencias en Inmunología, Escuela Nacional de Ciencias Biológicas, Instituto Politécnico Nacional (ENCB-IPN), Mexico City 11340, Mexico; miguelpradyu1401@gmail.com; 2Departamento de Fisiología, Instituto Nacional de Cardiología Ignacio Chávez, Mexico City 14080, Mexico; fausto22@yahoo.com; 3Departamento de Nefrología, Hospital Angeles Puebla, Mexico City 72190, Mexico; ivanquiroz621@gmail.com; 4Doctorado en Ciencias Biológicas y de la Salud, Universidad Autónoma Metropolitana, Mexico City 14387, Mexico; gmcarlos93@gmail.com; 5Departamento de Fisiopatología Cardio-Renal, Instituto Nacional de Cardiología Ignacio Chávez, Mexico City 14080, Mexico; horace_33@yahoo.com.mx; 6Departamento de Nefrología, Hospital General de Zona No. 18, Instituto Mexicano del Seguro Social, Playa del Carmen 77710, Mexico; daniel_00_5@hotmail.com; 7Departamento de Patología, Instituto Nacional de Cardiología Ignacio Chávez, Mexico City 14080, Mexico; virgiliasoto@gmail.com; 8Departamento de Nefrología, Instituto Nacional de Cardiología Ignacio Chávez, Mexico City 14080, Mexico; bernardomoguel@hotmail.com; 9Departamento de Inmunología, Escuela Nacional de Ciencias Biológicas, Instituto Politécnico Nacional (ENCB-IPN), Mexico City 11340, Mexico; rchacons@ipn.mx; 10Departamento de Inmunología, Instituto Nacional de Cardiología Ignacio Chávez, Mexico City 14080, Mexico

**Keywords:** antibody-mediated rejection, matrix metalloproteases (MMPs), tissue inhibitors of metalloproteases (TIMPs), renal graft dysfunction, biomarkers

## Abstract

Antibody-mediated rejection (ABMR) remains a major cause of renal graft dysfunction and loss. The histological hallmark of antibody-mediated rejection is progressive tissue damage, in which extracellular matrix turnover plays an important role. This turnover is mainly regulated by matrix metalloproteinases (MMPs) and tissue inhibitors of metalloproteinases (TIMPs). Recent studies suggest that MMP/TIMP imbalance may favor the progression of renal damage, inflammation, and fibrosis, but the utility of these molecules as a biomarker of antibody-mediated turnover has not been fully explored. We measured plasma MMP and TIMP levels by ELISA in 15 patients with antibody-mediated renal transplant rejection and 12 patients without rejection. There was a significant increase in MMP-1, MMP-2, and MMP-3 concentrations in the plasma of patients with rejection, directly correlating with the severity of different renal lesions. In contrast, TIMP-3 levels were elevated in patients without rejection, showing a negative correlation with the severity of histopathological lesions. The concentrations of these molecules demonstrated good diagnostic capacity for patients with rejection. Our results show that MMP-1, MMP-2, MMP-3, and TIMP-3 could be potential biomarkers of rejection.

## 1. Introduction

Acute antibody-mediated rejection (ABMR) remains one of the major challenges to renal transplant survival, as it contributes significantly to graft dysfunction and loss [1] ABMR is characterized by endothelial damage caused by the presence of anti-HLA antibodies, which promote the infiltration of T lymphocytes and macrophages into the allograft. This immune response triggers inflammation that leads to the destruction of renal structures, a process that involves degradation of the extracellular matrix (ECM) [2]. The maintenance of ECM homeostasis is primarily regulated by matrix metalloproteinases (MMPs), a family of zinc-dependent endopeptidases and tissue inhibitors of metalloproteinases (TIMPs) [3].

MMPs play a crucial role in physiological processes such as cell migration, lymphocyte invasion and immune modulation through the degradation of cytokines and chemokines. However, an imbalance between MMPs and TIMPs has been implicated in different pathological conditions, including tissue destruction, chronic inflammation, and fibrosis [4]. Recent studies suggest that dysregulated MMP/TIMP activity contributes to the progression of kidney diseases such as chronic kidney disease (CKD), glomerular disorders, and transplant rejection, frequently associated with histopathological features such as fibrosis [5].

Currently, renal biopsy remains the gold standard for diagnosing rejection, often supplemented by imaging studies and serum creatinine measurements. However, biopsies are invasive, costly, and prone to sampling errors, underscoring the need for non-invasive biomarkers that allow the early detection of ABMR [6]. In recent years, there has been an emerging interest in MMPs and TIMPs as promising biomarkers in different renal pathologies and transplant rejection. Particularly, MMP-1, MMP-2, MMP-3, MMP-9, TIMP-1, and TIMP-3 have been implicated in inflammatory process, ECM remodeling, and the development of histological alterations, including interstitial fibrosis [7].

Previous studies have provided evidence supporting their role in transplant rejection. For instance, Yan et al. [8] reported a significant increase in MMP-2 and TIMP-1 expression in kidney biopsies from patients with chronic humoral rejection, correlating with increased renal dysfunction. Other studies have confirmed the elevated levels of MMP-2 in urine samples from patients with chronic humoral rejection [9]. Furthermore, MMP-3 has been shown to increase significantly during ABMR episodes, distinguishing between severe and non-severe rejection [10]. In addition, excessive MMP-9 and MMP-2 activity has been associated with CKD progression by facilitating ECM degradation, increasing collagen turnover, and potentially aggravating renal allograft damage [11]. Conversely, TIMPs play a protective role by mitigating inflammation processes and the development of fibrotic histopathological changes [12]. Studies in murine models have shown that TIMP-3 deficiency (TIMP3^−/−^) results in the excessive deposition of type I collagen, tubular atrophy, and an increase in the severity of fibrotic interstitial lesions [13]. In patients, serum levels of TIMP-1 and TIMP-3 have been found to increase during the process of acute kidney injury, which has been suggested as a possible non-invasive marker of acute kidney injury [14]. Although MMPs and TIMPs are key regulators of extracellular matrix remodeling and have been linked to renal injury, their specific role in renal transplant rejection remains unclear. They modulate processes of tissue remodeling, immune infiltration, and the maintenance of graft integrity, suggesting that they may play an important role in transplant rejection [15]. However, there is still insufficient evidence to explore their role as non-invasive biomarkers for the diagnosis of renal rejection [16]. Therefore, the aim of this study was to evaluate MMP and TIMP concentrations in transplant recipients with acute antibody-mediated rejection.

## 2. Results

### 2.1. Demographic Characteristics of the Study Population

A total of 27 patients were included in this study. Of these, 12 patients were classified as belonging to the non-rejection (NR) group and 15 as belonging to the ABMR group. No significant differences were observed in gender distribution, with a slightly higher proportion of women in the NR group (66.7%) compared to the ABMR group (40%) (*p* = 0.16). However, age showed a statistically significant difference, being higher in the NR group (51 years, interquartile range (IQR) 37–59) compared to the ABMR group (34 years, IQR 31–42) (*p* = 0.03). The body mass index (BMI) was similar in both groups. Regarding the etiology of renal disease, the most frequent causes was “unknown” and membranous glomerulonephritis (MGN) in both groups. Diabetes was a less frequent cause of renal disease in the studied cohort, with only 2 patients (7.4%), both of which were in the NR group. The vintage of the renal allograft in the ABMR group (85 months, IQR 37.5–112) was higher than in the NR group (37.8 months, IQR 10.2–102), although this difference did not reach the level of statistical significance (*p* = 0.22). Likewise, no differences were found in the proportion of patients with cadaveric donors, with a distribution of 58.3% in NR and 40% in ABMR patients (*p* = 0.29). Finally, allosensitization, defined as the presence of preformed antibodies against the graft, was more prevalent in the ABMR group (73.3%) compared to NR (50%), although this difference did not reach the level of statistical significance (Table 1).

### 2.2. Histopathological Characteristics of Renal Allograft Biopsies

Patients with ABMR showed a higher frequency of inflammatory and chronic lesions compared to the group without rejection. Acute inflammatory lesions such as glomerulitis (g) were present in 100% (*n* = 15) of the patients with ABMR, compared to the NR group, where they were present in 48.3% (*n* = 7) of patients; the difference was statistically significant (*p* = 0.01). Similarly, peritubular capillaritis (ptc) was present in 100% of the cases with rejection, while it was present in only one of the patients without rejection, displaying significant difference (*p* < 0.001). Interstitial inflammation (i) was more frequent in the group with ABMR as it was absent in the group without rejection (60% vs. 0%, *p* = 0.001).

In relation to renal tubule lesions, tubulitis (t) remained present in 53.3% of patients with ABMR; in contrast, an absence of rejection was observed in 8.3% of the population, with the difference being significant (*p* = 0.02). Additionally, a marker of chronic damage, such as the presence of interstitial fibrosis (ci), was present in 40% of patients with ABMR and remained absent in the group that did not present rejection (Table 2).

### 2.3. Plasma MMP-1, MMP-2, and MMP-3 Levels in Patients with Antibody-Mediated Rejection (ABMR)

As shown in Figure 1, plasma concentrations of MMP-1, MMP-2, MMP-3, MMP-9, the MMP-9/TIMP-1 complex, TIMP-1, and TIMP-3 were analyzed. We found significantly higher plasma concentrations of MMP-1 (Figure 1a) in the ABMR group when compared to the NR group (medians: 1401 pg/mL vs. 949 pg/mL; *p* = 0.019). We also found a significant increase in MMP-2 (Figure 1b) in the ABMR group (3203 pg/mL vs. 2381 pg/mL; *p* = 0.025). Similarly, MMP-3 (Figure 1c) showed a significant difference (867.4 pg/mL vs. 565 pg/mL; *p* = 0.006). However, TIMP-3 (Figure 1g) levels were significantly higher in non-rejection (NR) patients compared to those with ABMR (mean: 1416 pg/mL vs. 677.8 pg/mL) (*p* = 0.030). In contrast, no significant differences were observed in the levels of MMP-9 (Figure 1d), TIMP-1 (Figure 1f), and the MMP-9/TIMP-1 complex (Figure 1e) between the NR and ABMR groups (*p* > 0.05).

### 2.4. Plasma Concentrations of MMP-1, MMP-2, MMP-3, TIMP-3 Can Predict ABMR

Subsequently, we evaluated the predictive capacity of MMP-1, MMP-2, MMP-3, and TIMP-3 by ROC curve analysis, revealing their significant diagnostic potential to discriminate between patients with and without renal transplant rejection. For the calculation of the odds ratio (OR), cutoff values were established for each MMP based on the Youden index: 1231.07 pg/mL for MMP-1, 3195.9 pg/mL for MMP-2, 656.74 pg/mL for MMP-3, and 987.75 pg/mL for TIMP-3. The optimal cutoff point for the combined TIMP-3/MMP-2 model was set at 0.23, whereas for the integrated model of MMP-1, MMP-2, and MMP-3, the optimal threshold was 0.49. In addition, the optimal cutoff point for the MMP-1/MMP-2 combination was 0.26, and for the MMP-1, MMP-3, and TIMP-3 combination, the threshold was 0.23. The values correspond to the probability coefficients generated by multivariate logistic regression.

MMP-1 showed an AUC of 0.763 (95% CI: 0.577–0.950), with a sensitivity of 60% and specificity of 100%, indicating a high positive predictive value (PPV = 90.16%) and an OR of 9.81, suggesting that patients with MMP-1 levels above 656.74 pg/mL have a higher risk of rejection. Similarly, MMP-2 presented an AUC of 0.762 (95% CI: 0.575–0.950), with a sensitivity of 53.8%, a specificity of 91.7%, and a PPV of 89.15%, leading to an OR of 6.28. Furthermore, MMP-3 exhibited the highest discriminatory capacity among individual biomarkers, with an AUC of 0.805 (95% CI: 0.632–0.979), a sensitivity of 80%, a specificity of 75%, a PPV of 80.28%, and an OR of 19.5 (Figure 2a).

In contrast, TIMP-3 showed an AUC of 0.744 (95% CI: 0.547–0.941), with a sensitivity of 75%, a specificity of 73.3%, a PPV of 78.16%, and an OR of 0.13, indicating a possible protective effect in terms of regulating rejection. However, given that TIMP-3 levels were lower in patients with ABMR, te true AUC was 0.234. To ensure a consistent comparison with other biomarkers, the ROC curve for TIMP-3 was normalized, aligning it with the interpretative framework used for MMPs while preserving its biological significance (Figure 2a).

To further assess the combined predictive capacity of these biomarkers, we employed multiple logistic regression analysis. This approach allowed us to integrate the individual contributions of MMP-1, MMP-2, MMP-3, and TIMP-3, optimizing the discrimination between patients with and without rejection. The combined biomarker analysis significantly improved diagnostic accuracy, with the combination of MMP-1, MMP-2, and MMP-3 demonstrating the highest predictive capacity, achieving an AUC of 0.967 (95% CI: 0.907–1), a sensitivity of 92.3%, specificity of 91.7%, a PPV of 93.40%, and an OR of 32.5. Additionally, the MMP-2/TIMP-3 combination reached an AUC of 0.894 (95% CI: 0.757–1), with a sensitivity of 80%, specificity of 100%, a PPV of 100%, and an OR of 71.4, further supporting the potential role of these biomarkers in distinguishing ABMR from non-rejection cases (Figure 2b).

The combined biomarker analysis significantly improved diagnostic accuracy. Specifically, the combination of MMP-1 and MMP-3 yielded an AUC of 0.916 (95% CI: 0.814–1), with a sensitivity of 100%, a specificity of 66%, a PPV of 79.24%, and an odds ratio (OR) of 329.37. Furthermore, the combination of MMP-1, MMP-3, and TIMP-3 achieved an even better diagnostic performance, with an AUC of 0.988 (95% CI: 0.969–1), a sensitivity of 100%, a specificity of 91%, a PPV of 93.85%, and an OR of 238 (Figure 2c,d).

### 2.5. Correlation of MMPs and TIMPs Concentrations with the Severity of Histopathological Findings

Moreover, we evaluated the correlation of plasma concentrations of MMPs with the severity of histopathological findings in renal biopsies. MMP-9/TIMP-1 complex concentrations showed a negative correlation with the presence of tubulitis (t) (Rho = 0.519, *p* = 0.007) (Figure 3a). TIMP-3 levels correlated negatively with the severity of glomerulitis (g) (Rho = −0.434, *p* = 0.023) (Figure 3b), but no significant correlation was observed with peritubular capillaritis (ptc). Conversely, MMP-1 concentrations were positively correlated with the severity of ptc (Rho = 0.411, *p* = 0.03), but not with g (Figure 3c). Notably, no significant correlations were found for MMP-2 with either ptc or g, despite its elevated plasma levels in patients with ABMR.

## 3. Discussion

In this study, we evaluated plasma concentrations of matrix metalloproteinases, including MMP-1, MMP-2, MMP-3, and the MMP-9/TIMP-1 complex, as well as tissue inhibitors of metalloproteinases, specifically TIMP-1 and TIMP-3, in patients with antibody-mediated renal allograft rejection. Our findings revealed significantly elevated levels of MMP-1, MMP-2, and MMP-3 in patients with ABMR compared with patients without rejection; conversely, TIMP-3 levels were decreased. Furthermore, these potential biomarkers demonstrated promising predictive ability for the diagnosis of ABMR, especially when combined as a panel. We also observed correlations between the plasma concentrations of these molecules and specific histopathological lesions characteristic of ABMR.

MMPs and TIMPs play crucial roles in maintaining extracellular matrix homeostasis through the regulation of degradation processes, angiogenesis, and tissue maintenance. The dysregulation of the balance between these molecules can promote inflammatory processes, tissue damage, fibrotic lesions, and vascular remodeling [5]. Previous research has explored the role of MMPs in the progression of various renal pathologies, including IgA nephropathy, kidney fibrosis, and chronic kidney disease progression [17]. Among the different types of MMPs, MMP-1, MMP-2, and MMP-3 have been identified as particularly important in the progression of kidney damage [18].

MMP-1, a collagenase that degrades structural components of the extracellular matrix, such as collagen type I and III, is essential for the maintenance of the kidney tubular structure [19]. Our results demonstrated a significant increase in plasma MMP-1 concentrations in patients with ABMR, a finding consistent with what has been reported in other studies, which reinforces its potential relevance. For example, in experimental studies of Adriamycin-induced nephropathy, histological cohorts were analyzed, and increased MMP-1 expression was found compared to controls, which could promote ECM accumulation and kidney damage [20,21]. However, diabetic mice showed the downregulation of MMP-1 [20,22]. Another study measured MMPs in their inactive form (zymogen) in patients with acute rejection and found an increase in proMMP-1 [5]. This elevation suggests that MMP-1 may exacerbate inflammatory processes in ABMR by facilitating the infiltration of immune cells into the graft, thereby promoting kidney dysfunction [23]. The modulation of MMPs activity emerges as a potential approach to regulate the immune response and improve graft survival, which reinforces the importance of continuing to investigate the impact of these enzymes in the context of kidney transplantation.

On the other hand, MMP-2, a gelatinase that degrades type IV collagen, is essential for basement membrane homeostasis and has been identified as a marker of kidney damage in models of CKD. MMP-2 plays a central role in the development of histopathological alterations by degrading ECM, leading to the release and activation of transforming growth factor-β (TGF-β), which induces fibroblast-to-myofibroblast differentiation and epithelial–mesenchymal transition (EMT), promoting excessive ECM accumulation and progression of kidney damage [24]. Our study showed higher levels of MMP-2 in patients with ABMR compared to patients with NR. For instance, a study by Wong et al. [9] showed increased MMP-2 levels in urine samples from patients with chronic humoral rejection, associated with an increase in proteinuria.

In addition, proMMP-2 and proMMP-3 levels increased in patients with chronic allograft nephropathy, which was related to decreased kidney function [25]. However, in our study, the levels of MMP-2 in its active form did not show a significant correlation with kidney function parameters such as serum creatinine or estimated glomerular filtration rate (eGFR). This finding suggests that the elevation of MMP-2 in ABMR is not directly associated with kidney functional deterioration but could reflect alterations in the remodeling of the ECM, favoring the infiltration of leukocytes in the graft and contributing to the inflammatory process [26]. In fact, studies in ischemia–reperfusion models have shown that MMP-2 inhibition reduces inflammation and preserves kidney function, as a significant reduction in tubular necrosis and the maintenance of kidney function was observed in MMP-2 knockout (−/−) mice [27]. This suggests that MMP-2 may play a crucial role in the pathogenesis of ABMR, mediating both inflammatory and tissue remodeling processes.

In addition, MMP-3, a stromelysin with the ability to degrade multiple structural components of the ECM, is crucial for the maintenance of kidney structure and function [28]. Previous studies have reported that MMP-1 and MMP-3 expressions correlate negatively with eGFR and positively with serum creatinine, suggesting their association with graft dysfunction [29]. In other studies, elevated MMP-3 levels were observed in patients with acute rejection; based on ROC curve analysis, the elevated MMP-3 levels showed a good predictive ability [10]. However, in contrast to what has been reported in other kidney pathologies, in our study, the increase in MMP-3 was accompanied by a simultaneous increase in MMP-1 and MMP-2. This finding suggests a possible distinctive pattern of MMP dysregulation in the context of ABMR, which could represent a specific molecular signature of acute antibody-mediated rejection. Additionally, MMP concentrations did not show a significant correlation with serum creatinine levels nor with the histological findings of the graft. This may suggest that the elevation of MMP-3 could be related to systemic inflammatory processes and alterations in extracellular matrix homeostasis, rather than being a direct reflection of kidney structural damage.

Furthermore, MMP concentrations did not show a significant correlation with s eGFR (Appendix A). This lack of association suggests that the elevation of MMP-1, MMP-2, and MMP-3 may reflect inflammatory processes and alterations in extracellular matrix homeostasis, rather than being directly related to impaired glomerular filtration.

In contrast, the main function of TIMPs is to inhibit the proteolytic activity of MMPs, maintaining the integrity of the ECM [30]. TIMP-3 is a broad-spectrum inhibitor and is most highly expressed in the kidney. Studies show that TIMP-3 has a protective effect against kidney damage by inhibiting chronic tubulointerstitial fibrosis, reducing the severity of tubular atrophy and cortical thinning, and increasing fibroblast activation and apoptosis [13,31]. TIMP-3 overexpression might indicate an attempt by the immune system to counteract tissue damage associated with MMP activation in non-rejection situations, suggesting a protective and modulatory role in kidney homeostasis. This finding is also relevant in the context of CKD, where an imbalance between MMPs and TIMPs has been associated with the development of renal fibrotic lesions.

Elevated levels of MMP-1, MMP-2, and MMP-3 in patients with ABMR could be related to the activation of exacerbated inflammatory responses, mediated by cytokines such as TNF-α and IL-1β, which induce the expression of these metalloproteinases and promote the degradation of the ECM [32]. This process facilitates immune cell infiltration and the structural disruption of kidney tissue, promoting rejection progression. In contrast, increased TIMP-3 in non-rejection patients suggests a potential protective mechanism, as this tissue inhibitor could limit uncontrolled proteolytic activity and preserve ECM integrity [33].

Furthermore, our analysis revealed associations between MMPs and TIMPs with the severity of histopathological lesions found in renal graft biopsies. The MMP-9/TIMP-1 complex was inversely correlated with the severity of tubulitis, which may indicate that an increase in this complex has the capacity to attenuate progressive tubular inflammation. Similarly, TIMP-3 concentrations demonstrated an inverse relationship with the intensity of glomerulitis, suggesting that inhibiting MMP activity may contribute to decreasing the inflammatory process in glomeruli. Conversely, MMP-1 levels showed a positive correlation with the severity of peritubular capillaritis, which may indicate that MMP-1 plays an important role in the pre-progression of microvascular inflammation. These findings may suggest that dysregulation between MMPs and TIMPs may favor the progression of allograft damage.

ROC analysis revealed that MMP-1, MMP-2, and MMP-3 present variable discriminatory ability for ABMR detection, with superior diagnostic performance compared to TIMP-3. However, the combination of biomarkers significantly improved diagnostic accuracy, highlighting the synergy between MMP-2 and TIMP-3, as well as the integrated panel of MMPs. In particular, the combination of MMP-1 and MMP-3 showed good performance, and when TIMP-3 (MMP-1/MMP-3/TIMP-3) was incorporated, the highest values of sensitivity, specificity, and AUC were achieved. These findings highlight the value of multivariate models versus the use of individual markers and reinforce the potential of these combinations as non-invasive diagnostic tools to identify episodes of rejection. On the other hand, other biomarkers such as miRNAs have shown good diagnostic capacity for ABMR, although they have variable results [34,35]. Unlike miRNAs, which mainly reflect changes in gene expression and post-transcriptional regulation, MMPs are actively involved in tissue damage, a function which could complement the assessment of rejection.

This study has several limitations that should be considered when interpreting these results. First, it has a relatively small size, which limits the statistical power and generalizability of the results. In addition, only one blood sample per patient, taken at the time of biopsy, was analyzed, which precludes the assessment of the temporal dynamics of biomarkers. Moreover, the lack of a healthy control group restricts the physiological interpretation of baseline MMP and TIMP levels. Also, urine samples and the analysis of their inactive forms of MMPs, which could have complemented the analysis as a non-invasive means of detection, were not included. The tissue expression of these molecules in biopsies was also not assessed, limiting the direct correlation between circulating levels and local production in the renal graft.

Finally, the age difference between the clinical groups could have influenced biomarker levels, and although age-stratified analyses were performed and no statistically significant differences were observed between the ABMR and NR groups (Appendix A), this could be a factor to consider when interpreting the results. Despite these limitations, the findings of this study are both interesting and provocative, offering novel insights into the potential utility of circulating MMPs and TIMPs as non-invasive biomarkers of acute antibody-mediated rejection. These preliminary results warrant further investigation and validation in larger, multicenter cohorts with longitudinal designs, multimodal biomarker assessments (blood, urine, tissue), and rigorous control of clinical and demographic variables.

## 4. Materials and Methods

### 4.1. Clinical Parameters

Patients with a history of kidney transplantation, who were treated at the “Ignacio Chávez” National Institute of Cardiology, were included in a cross-sectional study. Inclusion criteria included kidney transplant recipients older than 18 years, with eGFR ≥ 20 mL/min/1.73 m^2^ and a post-transplant time > 12 weeks, who underwent biopsy due to clinical indication or by protocol. Protocol biopsies were performed at three and twelvemonths post-transplant in patients with the presence of donor-specific antibodies (DSAs), detected pre-transplant by Panel Reactive Antibody—Single-Antigen (PRA-SA) using LABScreen^®^ Single Antigen, One Lambda, Thermo Fisher Scientific, Canoga Park, CA, USA, Complement-Fixing Crossmatch (CF-XM), or Complement-Dependent Cytotoxicity Crossmatch (CDC-XM)methods using LIFECODES^®^ Crossmatch Kit, Immucor, Norcross, GA, USA, in the absence of graft dysfunction. Patients with primary graft failure, neoplasia, active infection, or multiorgan transplantation were excluded. Additionally, those lacking DSA determination at the time of biopsy and those who did not have sufficient material for histopathological analysis or had incomplete clinical records were excluded. All participants provided informed consent in accordance with the Declaration of Helsinki, Good Clinical Practice guidelines, and local regulations. The study received approval from the local ethics committee under institutional protocol number: 24-1442.

### 4.2. Clinical-Pathological Diagnosis

The renal biopsies obtained were evaluated by a single nephropathologist following the criteria of the 2019 Banff meeting. C4d staining was performed on frozen tissue using immunohistochemistry (anti-human C4d antibody, Biomedica Gruppe, Vienna, Austria). The diagnosis of ABMR was established only when all three Banff diagnostic criteria were met. Biopsies considered as controls were those without evidence of rejection or specific pathological findings, classified as normal or showing nonspecific alterations according to the 2019 Banff guidelines.

### 4.3. Measurement of Circulating Anti-HLA Antibodies

All patients included presented a negative complement-dependent cytotoxicity crossmatch. Pre- and post-transplant anti-HLA antibodies were monitored in a single histocompatibility laboratory. Detection DSA analyses were performed against the HLA-A, -B, -C, -DRB1, -DRB345, -DQ and -DP loci using the Single-Antigen Flow Bead Assay, with measurement performed before transplantation and at the time of allograft biopsy. Patients with a mean fluorescence intensity (MFI) ≥ 1000 were considered positive, and de novo DSA (dnDSA) was considered positive when a donor-specific antibody with MFI ≥ 1000 was detected post-transplantation but was absent before transplantation.

### 4.4. Determination of MMPs and TIMPs Concentrations

Peripheral blood samples were obtained from renal transplant recipients at the time of allograft biopsy for the quantification of matrix metalloproteinases (MMP-1, MMP-2, MMP-3 and MMP-9) and tissue inhibitors of metalloproteinases (TIMP-1 and TIMP-3). Blood was drawn into tubes, with ethylenediaminetetraacetic acid (EDTA) as an anticoagulant (BD Vacutainer^®^, Becton, Dickinson and Company, Franklin Lakes, NJ, USA). The samples were immediately centrifuged at 2000 rpm for 5 min at room temperature. The resulting plasma was carefully separated and stored at −80 °C until further analysis. The quantification of these proteins was performed using a sandwich ELISA assay, following the manufacturer’s protocols (R&D Systems, Minneapolis, MN, USA). For this purpose, 96-well plates were used, which were coated with 100 µL of specific capture antibody and incubated at 4 °C for 24 h, followed by blocking with 1% BSA (Sigma-Aldrich, St. Louis, MO, USA) for 1 h at room temperature. Subsequently, 50 µL of plasma diluted 1:1 in assay buffer was added and incubated at 37 °C for 2 h. After washing with PBS-0.05% Tween-20 (Thermo Fisher Scientific, Waltham, MA, USA), 100 µL of specific biotinylated antibody was added and incubated for 1 h at room temperature, followed by the addition of 100 µL of streptavidin conjugated with peroxidase (HRP) (Thermo Fisher Scientific, Waltham, MA, USA), with incubation for 30 min. Detection was performed by adding 100 µL of TMB substrate (Invitrogen™, Thermo Fisher Scientific, Waltham, MA, USA) and incubating for 20 min in the dark; the reaction was stopped with 50 µL of 1 M sulfuric acid (Merck, Darmstadt, Germany). Absorbance was measured at 450 nm in a plate reader (Bio-Rad Model 680, Hercules, CA, USA), and the concentrations of each biomarker were calculated by interpolation on a standard curve.

### 4.5. Statistical Analysis

Statistical analysis was carried out through Statistical Package for Social Sciences SPSS version 26 (IBM Corp., Armonk, NY, USA), developed by IBM, Armonk, NY, USA. The Shapiro–Wilk test was used to determine the distribution of the data. In descriptive analysis, quantitative variables were represented as median and interquartile range values, and qualitative variables represented frequencies and percentages. Regarding to inferential analysis, the Mann–Whitney U test was used to compare the differences in quantitative variables between two groups of interest. Then, to evaluate the predictive capacity, the ROC (receiver operating characteristic) curve analysis was performed. The cutoff point was established using the Youden index. The chi-square test was used to analyze the qualitative variables. Finally, the Spearman test was used to perform the correlation analysis. A *p* < 0.05 was considered statistically significant.

## 5. Conclusions

In conclusion, our study shows that levels of MMP-1, MMP-2, MMP-3, and TIMP-3 have the potential to be non-invasive biomarkers in patients with acute antibody-mediated rejection. ROC curve analysis highlights the predictive capacity of these molecules, and their correlations with biopsy findings suggest their association with the severity of histologic ABMR alterations. However, further studies are needed to determine their relationship with long-term graft outcomes.

## Figures and Tables

**Figure 1 ijms-26-06011-f001:**
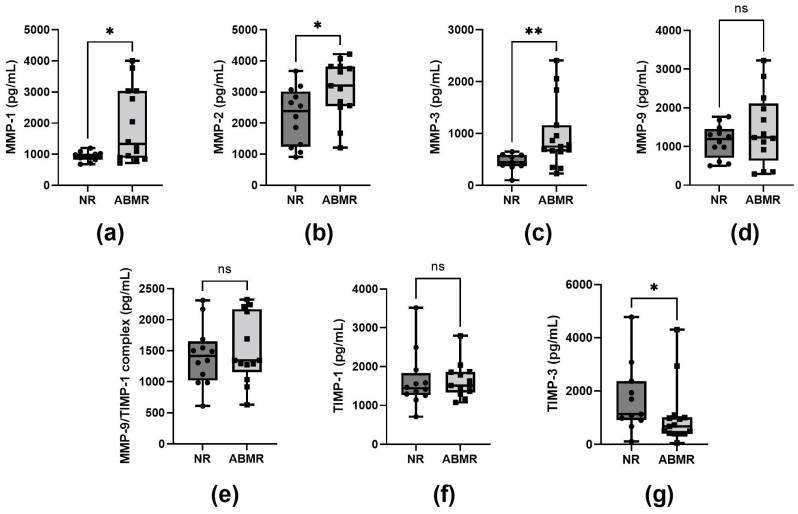
Analysis of plasma levels of matrix metalloproteinases (MMPs) and tissue inhibitors of metalloproteinases (TIMPs) in kidney transplant recipients with and without antibody-mediated rejection (ABMR). (**a**–**c**) MMP-1, MMP-2, and MMP-3 levels were significantly increased in ABMR patients compared with NR patients. (**g**) TIMP-3 levels were significantly higher in NR patients. (**d**–**f**) MMP-9, MMP-9/TIMP-1 complex, and TIMP-1 levels did not differ significantly between groups. Data are represented as box plots showing the median and interquartile range. Mann–Whitney U analysis was carried out. Values of *p* < 0.05 were considered significant. * *p* < 0.05, ** *p* < 0.01, ns = not significant.

**Figure 2 ijms-26-06011-f002:**
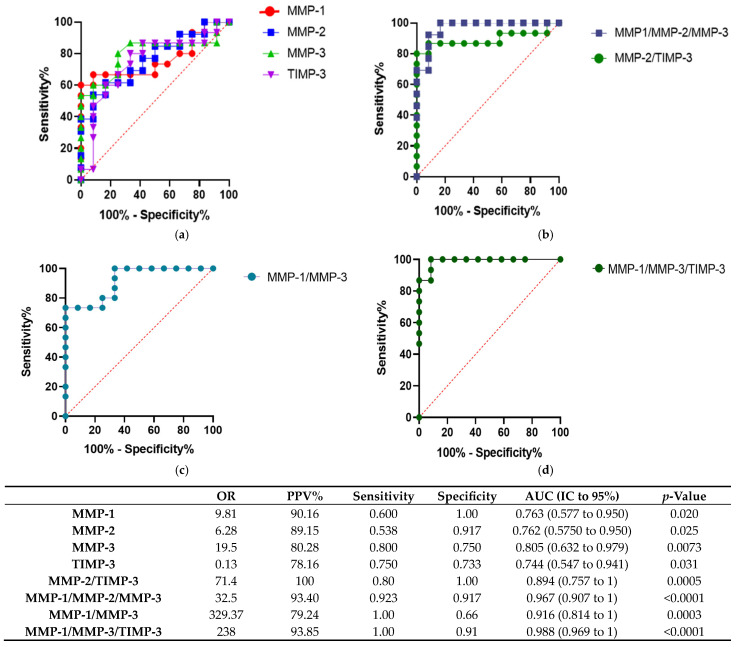
Analysis of discriminatory capacity of potential biomarkers and their combinations for detecting kidney transplant rejection using ROC curves. (**a**) ROC curves for individual biomarkers MMP-1, MMP-2, MMP-3, and TIMP-3. (**b**) ROC curves for combined models MMP-2/TIMP-3 and MMP-1/MMP-2/MMP-3. (**c**) Combined model including MMP-1 and MMP-3. (**d**) Combined model including MMP-1, MMP-3, and TIMP-3. The accompanying table presents the odds ratio (OR), positive predictive value (PPV%), sensitivity, specificity, and area under the ROC curve (AUC) with 95% confidence intervals (CI) for each marker and model. Since TIMP-3 levels were decreased in patients with rejection, the test variable was inverted prior to ROC analysis to maintain consistent directionality of the AUC. This transformation only affects the orientation of interpretation and does not alter the discriminative performance of the biomarker.

**Figure 3 ijms-26-06011-f003:**
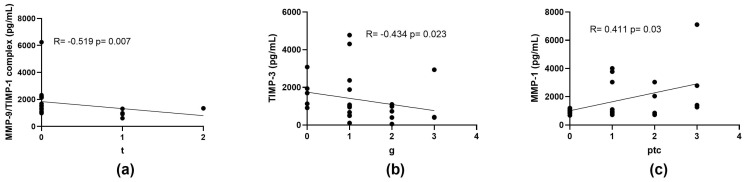
Spearman’s rank correlation between (**a**) MMP-9/TIMP-1 complex levels and tubulitis (t), (**b**) TIMP-3 levels and glomerulitis (g), and (**c**) MMP-1 levels and peritubular capillaritis (ptc). Each point represents individual sample. Linear regression lines are displayed with corresponding Spearman’s correlation coefficient (Rho) and *p*-value. Correlations were considered statistically significant at *p* < 0.05.

**Table 1 ijms-26-06011-t001:** Demographic characteristics.

	Total (*n* = 27)	NR (*n* = 12)	ABMR (*n* = 15)	*p* Value
**Female**	14 (51.9)	8 (66.7)	6 (40)	0.16
**Age**	42 (32–51)	51 (37–59)	34 (31–42)	0.03
**BMI (kg/m^2^)**	26.10 (21.90–29.30)	25.75 (21.85–29.85)	26.1 (21.95–28.2)	0.94
**Etiology**				0.12
**Unknown**	16 (59.3)	4 (41.6)	11 (73.3)	
**MGN**	9 (33.3)	5 (41.7)	4 (26.7)	
**Diabetes**	2 (7.4)	2 (16.7)	0	
**Time post-transplantation (months)**	80.5 (16–112)	37.8 (10.2–102)	85 (37.5–112)	0.22
**Deceased donor**	13 (48.2)	7 (58.3)	6 (40)	0.29
**Allosensitization**	17 (63)	6 (50)	11 (73.3)	0.2
**PRA I**	1 (0–5)	1 (0–2)	3 (0–10)	0.34
**PRA II**	2 (1–6)	2 (0.5–6)	2 (1–44)	0.57
**DSAs pre-transplantation**	9(33.3)	2 (16.7)	7 (46.7)	0.15
**DSAs post-transplantation**	20 (74.1)	6 (50)	15 (100)	0.003

Quantitative variables are expressed as median (interquartile range), and qualitative variables are expressed as absolute values (percentages). The *p*-value corresponds to the comparison between the non-rejection (NR) and acute antibody-mediated rejection (ABMR) groups. MGN: membranous glomerulonephritis; BMI: body mass index; PRA I/II: panel reactive antibodies class I and II; DSA: donor-specific antibodies.

**Table 2 ijms-26-06011-t002:** Classification of histopathological findings of renal biopsies according to Banff criteria.

	Total (*n* = 27)	NR (*n* = 12)	ABMR (*n* = 15)	*p*-Value
**g > 0**	22 (81.5)	7 (58.3)	15 (100)	0.01
**ptc > 0**	16 (59.3)	1 (8.3)	15 (100)	<0.001
**mm > 0**	22 (81.5)	7 (58.3)	15 (100)	0.01
**i > 0**	9 (33.3)	0	9 (60)	0.001
**t > 0**	9 (33.3)	1 (8.3)	8 (53.3)	0.02
**v > 0**	3 (11.1)	0	3 (20)	0.16
**cg > 0**	6 (22.2)	0	6 (40)	0.02
**ci > 0**	22 (81.5)	9 (75)	13 (86.7)	0.39
**ct > 0**	22 (81.5)	9 (75)	13 (86.7)	0.39
**cv > 0**	11 (40.7)	3 (25)	8 (53.3)	0.14
**i-IFTA > 0**	14 (52.9)	5 (41.7)	9 (60)	0.29
**C4d > 0**	9 (33.3)	5 (41.7)	4 (26.7)	0.34

Quantitative variables are expressed as median (interquartile range). Qualitative variables are expressed as absolute values (percentages). *p* value corresponds to the comparison between the no-rejection (NR) and antibody-mediated acute rejection (ABMR) groups. g: glomerulitis; ptc: peritubular capillaritis; mm: mesangial matrix expansion; i: interstitial inflammation; t: tubulitis; v: vasculitis (endarteritis); cg: transplant glomerulopathy; ci: interstitial fibrosis; ct: tubular atrophy; cv: vascular sclerosis; i-IFTA: inflammation in areas with interstitial fibrosis and tubular atrophy. C4d staining in peritubular capillaries (PTCs) by immunofluorescence/immunohistochemistry.

## Data Availability

The data will be made available at reasonable request to the corresponding author.

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
