# Peer review of "Kidney Transplant Recipients with Acute Antibody-Mediated Rejection Show Altered Levels of Matrix Metalloproteinases and Their Inhibitors: Evaluation of Circulating MMP and TIMP Profiles"

_ijms, 2025, doi:10.3390/ijms26136011_

Round 1
Reviewer 1 Report
Comments and Suggestions for Authors
Authors in the manuscript titled “Kidney Transplant Recipients with Acute Antibody-Mediated Rejection Show Altered Levels of Matrix Metalloproteinases and Their Inhibitors: Evaluation of Circulating MMP and TIMP Profiles” evaluated the plasma concentration of proteins MMP 1,2,3,9 and TMP1,3 in plasma collected from 15 patients. The authors analysis appears to indicate that patients with AMBR vs NR conditions show significant differences in MMP1, MMP3 and TIMP3 levels in support of their claims for these proteins to be indicated as potential biomarkers for Acute antibody mediated rejection (ABMR).
The major criticism is with the sample size, demographics and potential limitation to certain population group. Another major criticism is about not having a healthy people control group that may have strengthened the data as whole. Particularly since age appears to be significant in demographic distribution and therefore response to the transplant and therefore the protein levels between AMBR and NR groups.
My other comments include:
Section 2.1: Major
In the patient distribution as mentioned in lines 102, age appears to be a significant factor, and the appears to correlate with allosensitization or pre-formed antibodies in these younger individual’s vs the older population. Therefore, testing the levels of the biomarkers in healthy individuals is necessary and may provide a keener insight into the significance of these proteins as potential biomarker for AMBR.
Section 2.2:
The results for differences in plasma concentrations for the MPP1 and 3 and TMP3 clearly show differences in statistical significance, however the levels of MMP-2, TIMP-3 appear to have much broader ranges in the current sample set. Please provide raw data for these tests or show the distribution of individual data points on the graphs.
Also how age affects these levels? Please sub-classify the biomarker levels according to age then compare between NR and AMBR condition as it may be that the younger population in the NR and AMBR may show stronger differences in the plasma levels, discuss this distribution and show the data in supplementary information.
Section 2.3:
Since the MMP2 and TMP3 level appear to have broader distribution in plasma levels in the current sample set. Please perform similar ROC and multiple regression analysis with MMP-1 and MMP-3 only and a combination of MMP-1,3 and TIMP3, discuss the new combination and present the data as part of main figures if the above combinations are better performers if not provide them in supplementary information.
Section 3:
Lines: 227 to 229, the authors indicate that an inactive form MMP-1 the pro MMP1 was elevated in patients with acute rejection. Could the authors could provide similar evidence here with current study population? How the proMMP form could serve as biomarkers?
Lines: 237 to 242, the authors point out the functions of MMP2 in that it was shown to elevate TGF beta levels and contributes to kidney damage. Can the authors provide similar data on the levels of TGF beta in the current patient population? And similarly to TNF- alpha and IL- 1beta levels?
Lines: 242 to 243, similar to the mentioned study, can the authors provide urine levels of these biomarkers?
Lines: 260 to 262 talks about the eGFR. The authors must show how the eGFR levels correlate with MMPs here in this study?
Author Response
Comments 1: Authors in the manuscript titled “Kidney Transplant Recipients with Acute Antibody-Mediated Rejection Show Altered Levels of Matrix Metalloproteinases and Their Inhibitors: Evaluation of Circulating MMP and TIMP Profiles” evaluated the plasma concentration of proteins MMP 1,2,3,9 and TMP1,3 in plasma collected from 15 patients. The authors analysis appears to indicate that patients with AMBR vs NR conditions show significant differences in MMP1, MMP3 and TIMP3 levels in support of their claims for these proteins to be indicated as potential biomarkers for Acute antibody mediated rejection (ABMR).
The major criticism is with the sample size, demographics and potential limitation to certain population group. Another major criticism is about not having a healthy people control group that may have strengthened the data as whole. Particularly since age appears to be significant in demographic distribution and therefore response to the transplant and therefore the protein levels between AMBR and NR groups.
My other comments include:
Section 2.1: Major
In the patient distribution as mentioned in lines 102, age appears to be a significant factor, and the appears to correlate with allosensitization or pre-formed antibodies in these younger individual’s vs the older population. Therefore, testing the levels of the biomarkers in healthy individuals is necessary and may provide a keener insight into the significance of these proteins as potential biomarker for AMBR.
Response 1:
We fully recognize the importance of incorporating a healthy control group to strengthen the interpretation and robustness of biomarker-based findings. This point is addressed in the Discussion section, starting on line 359, where we explain how the lack of biomarker data from healthy individuals limits our results.
Section 2.2:
The results for differences in plasma concentrations for the MPP1 and 3 and TMP3 clearly show differences in statistical significance, however the levels of MMP-2, TIMP-3 appear to have much broader ranges in the current sample set. Please provide raw data for these tests or show the distribution of individual data points on the graphs.
Response 2:
In response, we have revised the corresponding figure to include each participant's individual data points within the box and whisker plots, allowing for a clearer visualization of the data dispersion.
Also how age affects these levels? Please sub-classify the biomarker levels according to age then compare between NR and AMBR condition as it may be that the younger population in the NR and AMBR may show stronger differences in the plasma levels, discuss this distribution and show the data in supplementary information.
Response 3:
To address this point, we performed an additional stratified analysis based on age, dividing the cohort into two groups: patients younger than 40 years and those older than 40 years. Biomarker levels within each age group were compared between the AMR and non-rejection (NR) conditions. These results have been added to the supplementary material (Supplementary Table 1). Furthermore, the potential influence of age on biomarker expression and the limitations of our current dataset in this regard are discussed in the discussion section.
Section 2.3:
Since the MMP2 and TMP3 level appear to have broader distribution in plasma levels in the current sample set. Please perform similar ROC and multiple regression analysis with MMP-1 and MMP-3 only and a combination of MMP-1,3 and TIMP3, discuss the new combination and present the data as part of main figures if the above combinations are better performers if not provide them in supplementary information.
Response 4:
In response, we performed additional ROC and multiple regression analyses using the combinations of MMP-1 and MMP-3, as well as MMP-1, MMP-3, and TIMP-3. These analyses are shown in Figure 2c-d).
Section 3:
Lines: 227 to 229, the authors indicate that an inactive form MMP-1 the pro MMP1 was elevated in patients with acute rejection. Could the authors could provide similar evidence here with current study population? How the proMMP form could serve as biomarkers?
Response 4:
Unfortunately, in the present study we were unable to evaluate plasma levels of the inactive proform of MMP-1 (proMMP-1) due to the lack of available resources and specific reagents necessary for its detection.
Lines: 237 to 242, the authors point out the functions of MMP2 in that it was shown to elevate TGF beta levels and contributes to kidney damage. Can the authors provide similar data on the levels of TGF beta in the current patient population? And similarly to TNF- alpha and IL- 1beta levels?
Response 5:
Unfortunately, we were unable to measure TGF-β, TNF-α, and IL-1β levels in the current patient population due to resource limitations and the lack of availability of the necessary reagents at the time of sample processing. In the Discussion section, we propose their inclusion in future studies to better understand the molecular pathways involved in graft injury.
Lines: 242 to 243, similar to the mentioned study, can the authors provide urine levels of these biomarkers?
Response 6:
Unfortunately, we did not have access to urine samples at the time of blood draws, which limited our ability to assess urinary levels of the biomarkers studied. However, we fully recognize the potential value of assessing these molecules in urine as a noninvasive approach.
Lines: 260 to 262 talks about the eGFR. The authors must show how the eGFR levels correlate with MMPs here in this study?
Response 7:
We appreciate the reviewer’s comment regarding the relationship between eGFR and MMP/TIMP levels. In response, we performed a correlation analysis between eGFR and the plasma levels of MMP-1, MMP-2, MMP-3, and TIMP-3. The results of these analyses have been added as a supplementary figure (Supplementary Figure 1).
Reviewer 2 Report
Comments and Suggestions for Authors
Note is made of the fact that these are preliminary findings from a relatively small number of renal allograft recipients with ABMR. Plus, some limitations have been identified both with how the data have been presented in the Results section, along with parts of the Discussion section of the manuscript. These include-
1) Were all of the histopathological abnormalities that are mentioned in the results-ie the tubulitis, the glomerulitis, and the peritubular capillaritis identified in all of the biopsies from each of the 15 patients who were diagnosed with ABMR or not? Can the authors elaborate on this? Plus what was the actual extent of each of the histopathologic abnormalities-mild/moderate/severe? Because in the absence of any other data to support how pervasive or significant the histologic abnormalities were then this limits the significance of the findings from the Spearmans rank correlation in Figure 3. Note is also made of the fact that no attempt was made to undertake more in depth histologic examination as to whether there was any expression of these molecules in the renal biopsies (another limitation).
2) The taking of just one blood sample at the time of renal allograft biopsy also confounds the results obtained for the levels of the MMP and TIMPs. It is possible that these were close to the peak levels that might be potentially elicited but this remains to be confirmed via further prospective studies explicitly looking into the trajectories of each around the time of acute rejection episodes. This limitation needs to be expanded upon in the Discussion section, along with whether it would also be useful to look into the concentrations of each MMP/TIMP in the urine as well.
3) All of the limitations of this study need to be addressed in more detail in the Discussion section along with the implications for what types of prospective studies now need to be performed.
Author Response
Note is made of the fact that these are preliminary findings from a relatively small number of renal allograft recipients with ABMR. Plus, some limitations have been identified both with how the data have been presented in the Results section, along with parts of the Discussion section of the manuscript. These include-
1) Were all of the histopathological abnormalities that are mentioned in the results-ie the tubulitis, the glomerulitis, and the peritubular capillaritis identified in all of the biopsies from each of the 15 patients who were diagnosed with ABMR or not? Can the authors elaborate on this?
Response 1:
All histopathological abnormalities mentioned in the results section—specifically, tubulitis, glomerulitis, and peritubular capillaritis—were systematically evaluated and documented in the biopsies of all 15 patients diagnosed with ABMR. For greater clarity and detail, we have expanded the description of these findings in the revised manuscript. This information is now included in the Results section, in subsection 2.2, Histopathological Features of Renal Allograft Biopsies, where the histological features identified in each biopsy are presented.
Plus what was the actual extent of each of the histopathologic abnormalities-mild/moderate/severe? Because in the absence of any other data to support how pervasive or significant the histologic abnormalities were then this limits the significance of the findings from the Spearmans rank correlation in Figure 3. Note is also made of the fact that no attempt was made to undertake more in depth histologic examination as to whether there was any expression of these molecules in the renal biopsies (another limitation).
Response 2:
Yes, all patients with rejection have glomerulitis (g) and peritubular capillaritis (ptc), as this is a necessary criterion for establishing this diagnosis.
- Regarding the correlation, there are patients with g0 and ptc 0 because these graphs include all patients (with and without rejection). Additionally, there are patients without rejection who may have mild glomerulitis (g1) without peritubular capillaritis (ptc 0) or vice versa but who do not meet the rejection criteria according to the Banff criteria (g + ptc >=2).
- Regarding how many have mild, moderate, and severe cases, as we discussed, we would have to calculate the percentage of g1, g2, g3, and ptc1, ptc2, and ptc3.
2) The taking of just one blood sample at the time of renal allograft biopsy also confounds the results obtained for the levels of the MMP and TIMPs. It is possible that these were close to the peak levels that might be potentially elicited but this remains to be confirmed via further prospective studies explicitly looking into the trajectories of each around the time of acute rejection episodes. This limitation needs to be expanded upon in the Discussion section, along with whether it would also be useful to look into the concentrations of each MMP/TIMP in the urine as well.
Response 3:
We fully agree that the use of a single blood sample obtained at the time of renal allograft biopsy limits the ability to determine the observed levels of MMPs and TIMPs. This limitation has been discussed in more detail in the Discussion section.
3) All of the limitations of this study need to be addressed in more detail in the Discussion section along with the implications for what types of prospective studies now need to be performed.
Response 4:
In response, we have expanded the Discussion section to provide a more detailed discussion of the limitations of the study, including the sample size, cross-sectional design, lack of longitudinal sampling, and lack of additional clinical parameters such as urinary biomarkers.
Reviewer 3 Report
Comments and Suggestions for Authors
Authors present a study in which they analyzed the potential role of MMPs/TIMPs in ABMR diagnosis. This is another piece of information, that may help to improve ABMR diagnosis. However some questions remain to be answered:
1) I suggest to re-read the manuscript, Authors use ,renal fibrosis’ as a disease (L65 and later), this is a histological finding,
2) many abbreviations are not explained when used for the first time (e.g. L105, L320),
3) Figure 2: what are the values shown on Y axis? Can you please provide Y axis title?
4) Figure 3 can not be seen appropriately,
5) I suggest to change sentence in L245-247,
6) Materials and methods: if this was a retrospective analysis, how did you collect the blood for MMP/TIMP analysis? How patients agreed to collect their blood?
7) Please correct the sentence in L324-327,
8) I suggest to add limitations of this study, you did an analysis on very small group of patients,
9) I suggest to change the conclusions(L373-375), you’ve analyzed ABMR, not the progression of graft damage, some Authors do not relate ABMR with the risk of graft failure.
Author Response
Authors present a study in which they analyzed the potential role of MMPs/TIMPs in ABMR diagnosis. This is another piece of information, that may help to improve ABMR diagnosis. However some questions remain to be answered:
1) I suggest to re-read the manuscript, Authors use ,renal fibrosis’ as a disease (L65 and later), this is a histological finding,
Response 1:
We thank the reviewer for this important observation. In response to the suggestion, we have revised the terminology throughout the manuscript, starting from line 65, to correctly refer to renal fibrosis as a histopathological finding rather than a disease entity. This change was made to ensure scientific accuracy and to align with proper pathological nomenclature. We appreciate the reviewer’s attention to detail in helping us improve the clarity and precision of the manuscript.
2) many abbreviations are not explained when used for the first time (e.g. L105, L320),
Response 2.
Thank you for your comment. We have revised the manuscript and added the corresponding definitions for the abbreviations when first mentioned to improve clarity and facilitate reader understanding.
3) Figure 2: what are the values shown on Y axis? Can you please provide Y axis title?
Response 3:
Thank you for your observation. The values shown on the Y axis correspond to sensitivity. We have now included the appropriate Y axis title in the illustrations of Figure 2 to clarify this information for the reader.
4) Figure 3 can not be seen appropriately,
Response 4:
Thank you for your feedback. We have adjusted the layout of Figure 3 to improve its visualization and ensure that all elements can be clearly observed and interpreted.
5) I suggest to change sentence in L245-247,
Response 5:
The sentence originally found between lines 245–247 has been revised for improved clarity and scientific accuracy. The updated version can now be found between lines 282 and 284.
6) Materials and methods: if this was a retrospective analysis, how did you collect the blood for MMP/TIMP analysis? How patients agreed to collect their blood?
Response 6:
We would like to clarify that the study was conducted as a cross-sectional analysis, not a retrospective one. The use of the term "retrospective" in the initial version was a mislabeling during the early drafting of the manuscript, and we have corrected this throughout the text for consistency.
All blood samples for MMP/TIMP analysis were collected at the time of clinically indicated or protocol renal allograft biopsies, after obtaining informed consent from each participant in accordance with institutional ethical guidelines.
7) Please correct the sentence in L324-327,
Response 7:
Thank you for your observation. We have revised the sentence between lines 324–327 to improve its clarity and scientific accuracy. The changes were made to better reflect the intended meaning and ensure consistency with the rest of the manuscript.
8) I suggest to add limitations of this study, you did an analysis on very small group of patients,
Response 8:
We have addressed the limitations of our study, including the small sample size, in the Discussion section. This acknowledgment highlights the need for future studies with larger cohorts to validate our findings and improve their generalizability.
9) I suggest to change the conclusions(L373-375), you’ve analyzed ABMR, not the progression of graft damage, some Authors do not relate ABMR with the risk of graft failure.
Response 9:
Thank you for your valuable observation. We have revised the conclusion section (lines 373–375) to more accurately reflect the scope of our study, focusing specifically on the association of the biomarkers with ABMR rather than with the progression of graft damage. The modification was made based on your suggestion to ensure the interpretation remains aligned with the presented data.
Round 2
Reviewer 2 Report
Comments and Suggestions for Authors
Note has been made of the revisions to the manuscript that have been made by the authors in response to the reviewers comments
Author Response
Thank you for the review
Reviewer 3 Report
Comments and Suggestions for Authors
Thank You very much for improving the manuscript.
Author Response
Thank you for your review